# Chemical Bonding and Physical Properties in Quasicrystals and Their Related Approximant Phases: Known Facts and Current Perspectives

Enrique Maciá Barber

Departamento Física de Materiales, Facultad CC. Físicas, Universidad Complutense de Madrid, 28040 Madrid, Spain; emaciaba@ucm.es; Tel.: +34-91-394-4745

**Featured Application: Thermoelectric materials.**

**Abstract:** Quasicrystals are a class of ordered solids made of typical metallic atoms but they do not exhibit the physical properties that usually signal the presence of metallic bonding, and their electrical and thermal transport properties resemble a more semiconductor-like than metallic character. In this paper I first review a number of experimental results and numerical simulations suggesting that the origin of the unusual properties of these compounds can be traced back to two main features. For one thing, we have the formation of covalent bonds among certain atoms grouped into clusters at a local scale. Thus, the nature of chemical bonding among certain constituent atoms should play a significant role in the onset of non-metallic physical properties of quasicrystals bearing transition-metal elements. On the other hand, the self-similar symmetry of the underlying structure gives rise to the presence of an extended chemical bonding network due to a hierarchical nesting of clusters. This novel structural design leads to the existence of quite diverse wave functions, whose transmission characteristics range from extended to almost localized ones. Finally, the potential of quasicrystals as thermoelectric materials is discussed on the basis of their specific transport properties.

**Keywords:** quasicrystals; chemical bond; thermoelectric materials

## 1. Introduction

Daniel Shechtman was awarded the Nobel Prize in Chemistry 2011 "for the discovery of quasicrystals", a novel phase of matter first reported in an Al-Mn alloy on 12 November 1984 [1], characterized by the presence of a quasiperiodic atomic long-range order [2], along with point group symmetries which are not allowed by the classical crystallography restriction theorem, namely, icosahedral, octagonal, decagonal, or dodecagonal ones. It should be highlighted that the term quasicrystal (QC) is just a shorthand for "quasiperiodic crystal", envisioned as a natural extension of the classical crystal notion, now embodying quasiperiodic arrangements of matter as well [2]. Therefore, this class of solids has nothing to do with amorphous or disordered materials [3–6].

Albeit the Al-Mn alloy discovered by Shechtman was thermodynamically unstable, transforming into a periodic crystal phase upon heat treatment, the existence of thermodynamically-stable QCs was subsequently reported in the Al-Li-Cu and Zn-Mg-Ga alloy systems in 1986 and 1987, respectively [7,8]. Currently, more than sixty different QC compounds, belonging to about twenty different alloy systems, have been reported to be thermodynamically-stable up to their respective melting points and to display Bragg reflections of extraordinary quality (Figure 1a), comparable to those observed for the best monocrystalline samples ever grown [4,5,9].

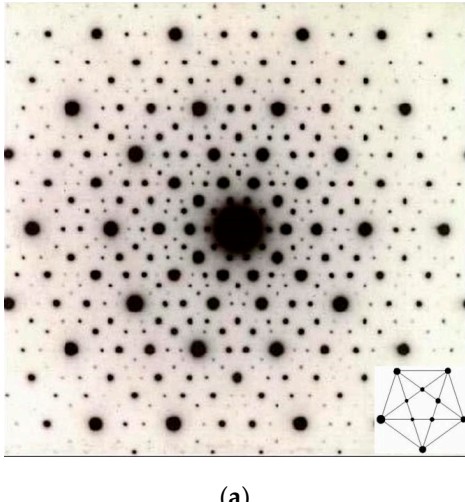
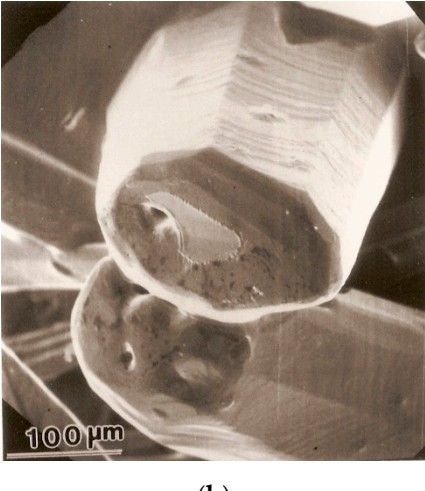

(**a**)　　　　(**b**)

**Figure 1.** (**a**) Electron diffraction pattern of the d-$Al_{70}Ni_{15}Co_{15}$ quasicrystal (QC) showing a ten-fold symmetry axis around the origin and a dense distribution of Bragg reflections. A set of consecutive nested pentagons, exhibiting the Pythagorean pentagram self-similar design, depicted at the bottom right corner, can be clearly seen; (**b**) scanning electron image showing the decagonal prism habit growth morphology corresponding to a d-AlNiCo QC. Cross sections of the decaprism show ten faceted planes where atoms are quasiperiodically arranged. These quasiperiodic planes are periodically stacked along the prism axis (Courtesy of An Pang Tsai).

The growth of relatively large, high structural quality samples, exhibiting pentagonal dodecahedron or triacontahedral shapes in the case of icosahedral symmetry QCs (usually denoted i-QCs) and octagonal, dodecagonal, or decagonal prismatic habits (Figure 1b) in the case of axial symmetry QCs (denoted o-, dd-, and d-QCs, respectively), allowed for the detailed experimental study of intrinsic physical properties of QCs, including their electrical, thermal, magnetic, optical, and mechanical properties. From the experimental results collected during the last thirty years on the physical properties of stable QCs one concludes that these intermetallic compounds certainly possess a remarkable set of unique features, which significantly differ from those observed in their periodic counterparts [3,6,10]. Indeed, metallic substances show a number of characteristic physical attributes stemming from the presence of a specific kind of chemical bond among their atomic constituents: the so-called metallic bond [11].

In Table 1 we list a number of representative physical properties of both typical metals and QCs showing that quasicrystalline alloys significantly depart from standard metallic behavior. For instance, in Figure 2 we see that the electrical conductivity of samples belonging to different alloy systems steadily increases as the temperature increases (up to values close to the melting point, see Section 6). In addition, their room temperature electrical conductivity values take on remarkably low values (50–400 $\Omega^{-1}cm^{-1}$), as compared to those usually reported for ternary alloys (3000–5000 $\Omega^{-1}cm^{-1}$), and these values very sensitively depend on minor variations of the sample stoichiometry, as it can be seen by inspecting Figure 2a and the inset of Figure 2b. Thus, the electrical transport properties of thermodynamically-stable i-QCs of high structural quality resemble a more semiconductor-like than metallic character. From the main frame of Figure 2b we see that the conductivity curves of different QC samples are nearly parallel up to about 1000 K, a feature referred to as the inverse Matthiessen rule, since one can write $\sigma(T) = \sigma(0) + \Delta\sigma(T)$, where $\sigma(0)$ measures the sample dependent residual conductivity, and $\Delta\sigma(T)$ is proposed to be a general function [12,13]. Therefore, QCs provide an intriguing example of ordered solids made of typical metallic atoms which do not exhibit the physical properties usually related to the presence of metallic bonding. Is this unusual behavior uniquely related to the characteristic long-range quasiperiodic order (QPO) present in the underlying atomic structure of QCs?

**Table 1.** Comparison between the physical properties of typical metallic compounds and QC alloys.

| Property | Metals | Quasicrystals |
|---|---|---|
| Electrical | High conductivity<br>Resistivity increases with temperature<br>Small thermopower | Low conductivity<br>Resistivity decreases with temperature<br>Moderate thermopower |
| Thermal | High conductivity<br>Large specific heat | Very low conductivity<br>Small specific heat |
| Magnetic | Paramagnetic<br>Ferromagnetic | Diamagnetic |
| Optical | Drude peak | No Drude peak |
| Mechanical | Ductility, malleability | Brittle |
| Tribological | Relatively soft<br>Moderate friction<br>Easy corrosion | Very hard<br>Low friction coefficients<br>Corrosion resistant |

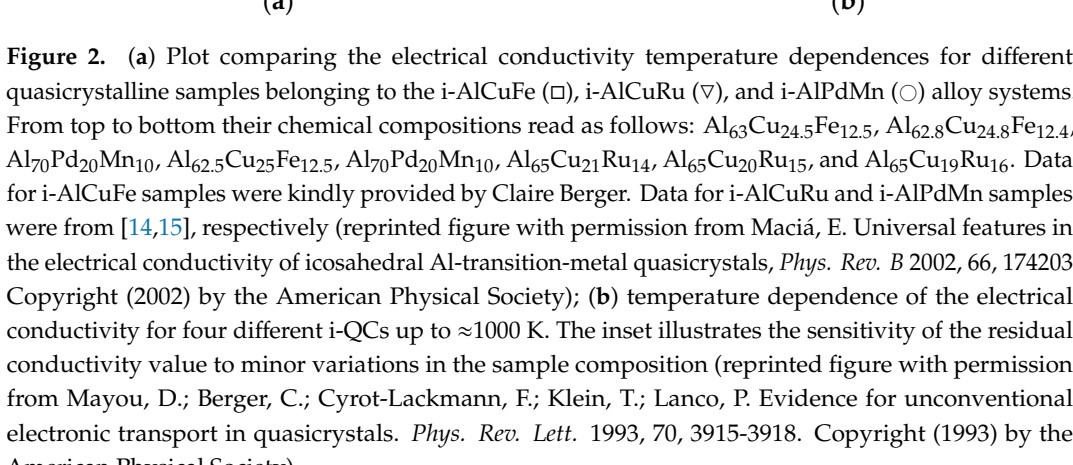

(a)                                    (b)

**Figure 2.** (**a**) Plot comparing the electrical conductivity temperature dependences for different quasicrystalline samples belonging to the i-AlCuFe ($\square$), i-AlCuRu ($\triangledown$), and i-AlPdMn ($\bigcirc$) alloy systems. From top to bottom their chemical compositions read as follows: $Al_{63}Cu_{24.5}Fe_{12.5}$, $Al_{62.8}Cu_{24.8}Fe_{12.4}$, $Al_{70}Pd_{20}Mn_{10}$, $Al_{62.5}Cu_{25}Fe_{12.5}$, $Al_{70}Pd_{20}Mn_{10}$, $Al_{65}Cu_{21}Ru_{14}$, $Al_{65}Cu_{20}Ru_{15}$, and $Al_{65}Cu_{19}Ru_{16}$. Data for i-AlCuFe samples were kindly provided by Claire Berger. Data for i-AlCuRu and i-AlPdMn samples were from [14,15], respectively (reprinted figure with permission from Maciá, E. Universal features in the electrical conductivity of icosahedral Al-transition-metal quasicrystals, *Phys. Rev. B* 2002, 66, 174203 Copyright (2002) by the American Physical Society); (**b**) temperature dependence of the electrical conductivity for four different i-QCs up to ≈1000 K. The inset illustrates the sensitivity of the residual conductivity value to minor variations in the sample composition (reprinted figure with permission from Mayou, D.; Berger, C.; Cyrot-Lackmann, F.; Klein, T.; Lanco, P. Evidence for unconventional electronic transport in quasicrystals. *Phys. Rev. Lett.* 1993, 70, 3915-3918. Copyright (1993) by the American Physical Society).

It is certainly tempting to think of these properties as the fingerprints of QPO in condensed matter. An assumption which is apparently supported by the observation of remarkable anisotropies in the transport properties of d-QCs, which behave as metals when measurements are performed along their periodic direction, whereas they exhibit a non-metallic behavior in the quasiperiodically ordered perpendicular planes [16]. Furthermore, their electrical conductivity significantly *decreases* as the structural quality of the sample is improved (e.g., by annealing), in striking contrast with periodic metallic alloys, whose electrical transport properties improve when structural imperfections are removed upon heating [12], thereby providing additional evidence on the role played by the novel kind of order present in these materials. Another suitable example is provided by the extremely low

thermal conductivity measured for QCs over the wide temperature interval 1–400 K [3,6,10], which could be explained in terms of an enhanced Umklapp process stemming from the dense filling of Bragg reflections in reciprocal space (Figure 1a).

Notwithstanding this, as far as I know no definitive theoretical explanation has been given to the physical origin of QCs' specific properties on the sole basis of the very nature of the new kind of underlying order they possess. On the contrary, a growing number of experimental results and numerical simulations alike suggest that a main factor in the origin of the remarkable properties of both d- and i-QCs is probably related to factors involving nearest and next-nearest atomic neighbors, thereby highlighting the important role of short-range effects in the emergence of unusual physical properties in QCs [17–20].

To gain some insight into this important issue it is convenient to note that, in most quasicrystalline forming alloy systems, the true QC phase is accompanied by a number of compositionally-related periodic crystals, having huge unit cell sizes (containing up to 150–200 atoms), which are called *approximant* phases, because these crystals not only have very similar compositions, but also atomic structures closely resembling that of the true QC compound, from which they can nevertheless be distinguished. Approximant crystals are important for understanding the structure of the related QCs since their atomic arrangements can be determined with high accuracy, which is very helpful in obtaining a first-approach QC structure model. Within the hyperspace formalism both periodic and QP structures can be derived by projection from a parent hyperlattice onto the 3D physical space [3,4], the main difference being that for QCs the projection involves irrational numbers, whereas in the case of approximant crystals the projection is given in terms of rational *approximant numbers* to those irrational quantities. Accordingly, approximant crystals to i-QCs can be classified by a rational order number $F_n/F_{n-1}$, which is a ratio of two consecutive terms in the Fibonacci sequence given by the recurrence formula $F_{n+1} = F_n + F_{n-1}$, with $F_0 = F_1 = 1$.

Thus, from a numerical study on covalent bonding and semiconducting bandgap formation in quasicrystalline approximants bearing Al and transition metal (TM) atoms, it was concluded that it is not the long-range QPO but an unusual Al-TM bonding that is responsible for a substantial part of the peculiarities observed in the transport properties of this QC alloy system [21]. A similar conclusion was reached from ab-initio studies of i-AlPdMn QCs suggesting that, since the long-range QPO modifies only to a limited extent the features in the electronic DOS resulting from the short-range order, then the local chemistry could be a more dominant factor in determining the electronic properties of these QCs [22].

Alternatively, the non-metallic behavior of QCs could also be understood as being due to the QP arrangement of atoms throughout the space, which naturally leads to the formation of a deep pseudogap near the Fermi energy, along with the existence of a significant number of electronic states exhibiting very low diffusion coefficients. To this end, one must take into account the existence of a richer behavior for electronic states in QCs: On the one hand, we have extended electronic wave functions (Figure 3a) able to open a pseudogap close to the Fermi level via diffraction and interference processes with quasiperiodically-stacked arrangements of atomic planes; on the other hand, we have localized electronic states as well (Figure 3b), stemming from resonant effects involving nested atomic clusters exhibiting self-similar geometries at different scales. This polyvalent transmission characteristic of electronic states in solids endowed with self-similar invariance symmetry is at the root of the so-called critical nature of these wave functions, which belong to fractal-like energy (vibration) spectra in the ideal case. Thus, critical electronic states embrace a diverse set of wave functions exhibiting a broad palette of possible diffusivity values, ranging from highly-conductive transparent states to highly-resistive, almost localized ones [23,24].

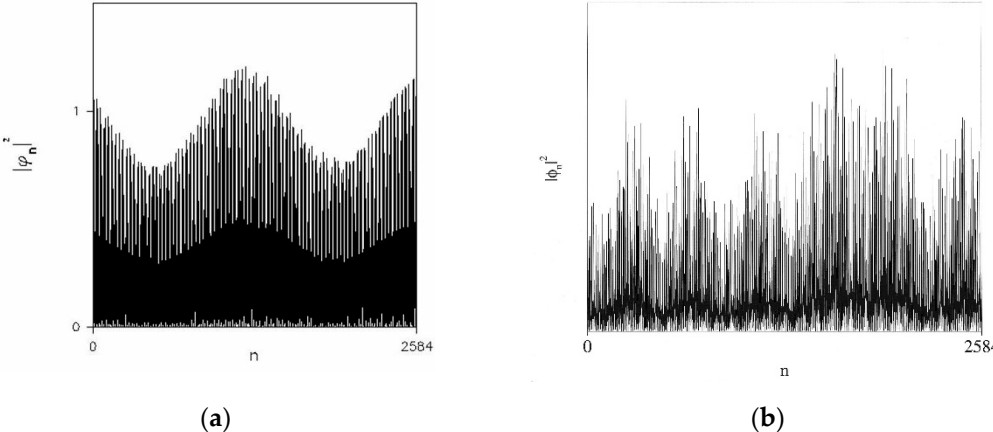

**Figure 3.** Two examples of different kinds of critical wave functions in one-dimensional Fibonacci QCs: (**a**) Electronic charge distribution for an extended state with E = −13/50 = −0.26 and a transmission coefficient T = 1 (reprinted from Maciá-Barber E. *Aperiodic Structures in Condensed Matter: Fundamentals and Applications;* CRC Press: Boca Raton, FL, 2009. With permission from CRC Taylor & Francis group); (**b**) electronic charge distribution for a low diffusivity state with E = −0.335440 . . . and a transmission coefficient T = 0.4689 . . .

## 2. The Hume-Rothery Electron Concentration Rule

The cohesive energy of a solid refers to the energy necessary to separate its constituent atoms from each other and bring them to an assembly of neutral free atoms. Therefore, in forming solids, cohesive energy is gained by lowering the total energy relative to that of the free atoms through the formation of different bonding types, namely, ionic, covalent, and metallic. Since ionic bonding originates from electrostatic interaction among charged atoms, it is not strictly necessary to consider the overlap of wave functions between neighboring atoms to evaluate the cohesive energy of these compounds. Conversely, in both covalent and metallic bonding types one must consider the formation of a valence band arising from quantum resonance effects involving the atomic orbitals of condensing atoms. Thus, as atoms get closer and closer to each other, the atomic orbitals in the original assembly of neutral atoms are lifted and split leading to the formation of a valence band upon hybridization. In this picture, the fundamental reason for the existence of the metallic state is that in an isolated metal atom the valence electrons occupy positions close to the upper edge of the potential well, so that the perturbations introduced during the condensation processes by neighboring metal atoms lead to delocalization of the valence electrons [25].

Accordingly, metallic bonding is described by considering that the free electrons charge is uniformly distributed throughout an ordered array of potentials due to positive ions and that the total charge of electrons exactly balances that of the ions. Along with the electrostatic energy among electrons and ions, one must include in the total energy budget the potential energy of electrons moving in the ionic lattice, the electron–electron interactions and the kinetic energy of electrons. Since no more than two electrons can occupy the same quantum state, the available electronic states are progressively filled up to the Fermi energy, $E_F$, and the resulting kinetic energy of the electronic system systematically increases with the electron concentration. Thus, in the free electron model the average kinetic energy per electron is given by $<T_e> = 3E_F/5 = 2903/(r_s/a_0)^2$ kJ mol$^{-1}$, where $r_s$ is the Fermi-surface radius and $a_0$ is the lattice parameter [26,27]. Hence, since $<T_e> > 0$, the larger the Fermi-surface size the more favorable the metallic bonding formation. Therefore, a variety of metallic phases are stabilized in nature by mechanisms able to lower the kinetic energy of electrons as much as possible. Of particular interest to us is the so-called Hume-Rothery concentration rule, since it was systematically exploited by Tsai and co-workers as a useful guide to synthesize the first generation of high-quality, thermodynamically-stable QCs mentioned in Section 1. This rule accounts for the

stabilization of certain alloys and compounds with closely related structures that exhibit the same ratio of number of valence electrons to number of atoms (the so-called electron-per-atom ratio, e/a) [27,28].

The Hume-Rothery rule can be explained as resulting from a perturbation of the kinetic energy of the valence electrons by their diffraction by the crystal lattice when an electron has such a wave length $\lambda = h(2mT_e)^{-1/2}$ and a direction so as to permit Bragg reflection from an important crystallographic plane. Due to this perturbation, a rearrangement of electronic states to both higher and, more importantly, lower energies take place. This leads to the occurrence of gaps or pseudogaps (that is, local deep minima) in the electronic density of states (DOS). Thus, special stability would be expected for metals with just the right number of electrons. In that case, $E_F$ lies close to a DOS minimum, so that only the states shifted to lower energies are occupied and the energy of the electronic system is decreased. This electron number is proportional to the volume of a polyhedron in the reciprocal space (the so-called Brillouin–Jones zone), corresponding to the crystallographic planes giving rise to the perturbation. In alloys with high enough symmetry, this zone is quite close to a spherical shape, and the diffraction condition can be conveniently expressed in the form $K_{hkl} = 2k_F$, where $K_{hkl}$ is the reciprocal vector of the considered diffraction plane, $k_F = (3\pi^2 n)^{1/3}/a_0$ is the radius of the Fermi sphere in reciprocal space, and $n = (e/a)N$ is the electron density, where N is the number of atoms in the unit cell.

A long-term study aimed at calculating the electron per atom ratio, e/a, in structurally-complex metallic alloys was undertaken by Mizutani and co-workers, who have shown that this magnitude plays a key role in the cohesion of diverse classes of solids and it is closely linked with the more usual valence concept. To this end, they exploited the so-called full-potential linearized augmented plane wave (FLAPW)-Fourier theory in order to study the electronic structure of many pseudogap-bearing binary and ternary intermetallic compounds, including Al-, Zn-, and Cd-based compounds, Zintl polar compounds, phosphorus-based compounds, and inter-transition metal compounds. In this way, reliable e/a values for 54 elements in the periodic table were obtained, including representatives from group 1 up to group 15 (Figure 4) [28]. Thus, the origin of a pseudogap at the Fermi level for a large number of materials has been successfully interpreted in terms of electronic interference conditions, regardless of the bond-types involved.

| 1 | 2 | | | | | | | | | | | 13 | 14 | 15 |
|---|---|---|---|---|---|---|---|---|---|---|---|----|----|----|
| Li 1.02 | Be 2.00 | | | | | | | | | | | B 2.98 | C 3.92 | N |
| Na 1.01 | Mg 2.01 | 3 | 4 | 5 | 6 | 7 | 8 | 9 | 10 | 11 | 12 | Al 3.01 | Si 4.00 | P 4.97 |
| K 1.01 | Ca 2.00 1.56 | Sc 2.94 1.33 | Ti 1.14 | V 0.90 | Cr 0.92 | Mn 1.05 | Fe 1.05 | Co 1.03 | Ni 1.16 | Cu 1.00 | Zn 2.04 | Ga 3.00 | Ge 4.05 | As 4.92 |
| Rb 1.01 | Sr 1.96 | Y 3.15 1.87 | Zr 1.49 | Nb 1.32 | Mo 1.39 | Tc 0.95 | Ru 1.04 | Rh 1.00 | Pd 0.96 | Ag 1.01 | Cd 2.03 | In 3.03 | Sn 3.97 | Sb 4.99 |
| Cs 1.04 | Ba 2.03 | La 3.00 | Hf 1.76 | Ta 1.57 | W 1.43 | Re 1.40 | Os 1.55 | Ir 1.60 | Pt 1.63 | Au 1.00 | Hg 2.03 | Tl 3.03 | Pb 4.00 | Bi 4.94 |

**Figure 4.** Electron per atom concentration ratio e/a for 54 elements in the periodic table. The value in the top (bottom) level for Ca, Sc, and Y is used when these atoms are alloyed with non-transition metal (transition metal) elements, respectively. (Reprinted from Mizutani U and Sato H, The physics of the Hume-Rothery electron concentration rule, Crystals 7, 9 (2017); doi:10.3390/cryst7010009, Creative Commons Attribution License CC BY 4.0).

The Hume-Rothery criterion can be applied to QCs by introducing a pseudo-Brillouin zone related to the most intense diffraction reflections. In doing so, we note that the high-order symmetry of i-QCs (as compared to periodic crystalline phases) generally provides an enhanced overlapping of the Fermi

surface with the pseudo-Brillouin zone in the reciprocal space. In fact, it has been confirmed that for QCs containing elements with a full d-band, such as i-$Zn_{43}Mg_{37}Ga_{20}$ (e/a = 2.221), i-$Al_{56}Li_{33}Cu_{11}$ (e/a = 2.129), i-$Zn_{60}Mg_{30}(RE)_{10}$ (e/a = 2.127, where RE stands for rare-earth atoms) or i-$Zn_{80}Sc_{15}Mg_5$ (e/a = 2.039), Fermi sphere–pseudo-Brillouin zone interaction gives rise to a significant reduction of the DOS (pseudogap) close to $E_F$, ultimately leading to the stabilization of these compounds as a result of distributing electrons with the highest kinetic energies into deeper states in the energy spectrum. Thus, the formation of a pseudogap at the Fermi level with a width of 0.5 to 1 eV and a height 0.2 to 0.6 times as high as the typical free electron DOS can lower the electronic energy by 30 to 50 kJ/mol [27,28].

On the other hand, in QCs bearing TM atoms, along with the Fermi-surface–Brillouin-zone interaction mechanism due to diffraction of delocalized electrons by ordered atomic lattice planes, the structure stabilization can be influenced by the wave functions overlapping among contiguous atoms, giving rise to the formation of bonding and antibonding levels due to orbitals hybridization between neighboring Al-3p and TM-3d atomic orbitals [27]. A detailed systematic investigation of Al-based i-QCs bearing TM elements, using soft X-ray emission spectroscopy, allowed for a quantitative analysis of the partial densities of 3p and 3s,d states at the aluminum edge of the electronic structure, as well as of 3d states at the TM edge, respectively [29]. From the obtained data hybridization between 3p and d states is clearly seen in all the considered samples (see Figure 9b), naturally arising the question regarding the relative importance of covalent bonding in the final stabilization of this class of QCs [30]. Additional evidence on the important role of orbital hybridization in QCs stability come from band structure calculations of i-Cd(Yb,Ca) QCs approximant crystals, indicating that low-lying d unoccupied states play an essential role in the formation of a pseudogap due to hybridization with Cd 5p states producing orbitals just below $E_F$. In this case, the Fermi energy is not located close to the bottom of the resulting pseudogap (as it is usual for QCs stabilized via the Fermi-surface–Jones-zone mechanism), but is pinned at a shoulder of the DOS [31]. Similar numerical studies for approximant crystals related to i-Ag-In-Yb QCs, which are obtained from i-Cd-Yb by replacing Cd by Ag and In (the adjacent elements to Cd in periodic table, thereby preserving the original overall e/a ratio), also yield the presence of a hybridization-induced pseudogap in the DOS close to the Fermi level. These theoretical results were, subsequently, experimentally confirmed from ultraviolet photoemission spectroscopy studies [32,33].

Therefore, one reasonably expects the nature of chemical bonding among certain constituent atoms should play a significant role in the onset of non-metallic physical properties of QCs bearing TM elements. Accordingly, any prospect on the origin of the pseudogap at the Fermi level in the Al-Cu-(Fe,Ru,Os), Al-Pd-(Mn,Re), and Al-Cu-(Ni,Co) alloy systems should consider both metallic and covalent bonding styles in order to determine the relative number of itinerant versus localized electrons in the resulting QCs. This issue will be further discussed in Section 6.

## 3. The Chemical Synthesis Route to New Quasicrystals

Elements composing thermodynamically-stable QCs found to date belong to the broad chemical family of metals, including representatives from alkali, alkaline earth, transition metals, or rare-earth blocks (Figure 5). By inspecting Figure 5 we also see that most main forming elements cluster in groups 11–13 and 4, whereas minority elements are mainly found among TM or RE elements. Thus, most metallic atoms are able to participate in the formation of quasicrystalline phases when one adopts the proper stoichiometric ratios. Indeed, as a matter of fact all of the stable QCs discovered to date have very narrow composition ranges in their respective phase diagrams, indicating that the alloy chemistry strongly affects the stability of these compounds. For instance, the minority atom constituent in the ternary systems

$$Al_{72}Pd_{17}\begin{pmatrix} Ru \\ Os \end{pmatrix}_{11}, \quad Al_{63}Cu_{25}\begin{pmatrix} Fe \\ Ru \\ Os \end{pmatrix}_{15}, \quad Al_{70}Pd_{20}\begin{pmatrix} Mn \\ Tc \\ Re \end{pmatrix}_{10},$$

and $Zn_{60}Mg_{30}(Y,Gd,Tb,Dy,Ho,Er)_{10}$, belong to the same group or row of the periodic table, hence indicating the importance of their electronic structure or ionic radii for the stability of the compound, respectively. Similar trends can be observed for the main forming elements as well. For instance, since Cd is located in the same column as Zn in the periodic table, the family of stable i-QCs $Cd_{65}Mg_{20}(Y,Gd,Tb,Dy,Ho,Er,Tm,Yb,Lu,)_{15}$, was obtained by replacing Zn with Cd in the parent i-$Zn_{50}Mg_{42}RE_8$ system. Note that the relative content of Mg significantly decreases in the Cd-Mg-RE compounds as compared to the Zn-Mg-RE ones. Since Mg atoms have the smallest ionic radius of all the involved atoms in these alloys, this stoichiometric change may be related to an optimization of the available space in the resulting compounds. Indeed, it is worth noticing that RE elements with atomic radii larger than 1.8 Å apparently do not form stable QCs in the Cd-Mg-RE and Zn-Mg-RE alloy systems, hence suggesting that there exists a restriction on the atomic size in this case. A parameter (referred to as the effective atomic size ratio), has been introduced in order to analyze the role of atomic size effects in QCs' stability. This parameter takes into account both atomic radius, $r_\alpha$, and concentration, $C_\alpha$, in the form $R_{r,e} = (r_M C_M)/(r_A C_A)$, where the subscript A indicates the major constituent, and M stands for secondary or minor elements in the alloy [33].

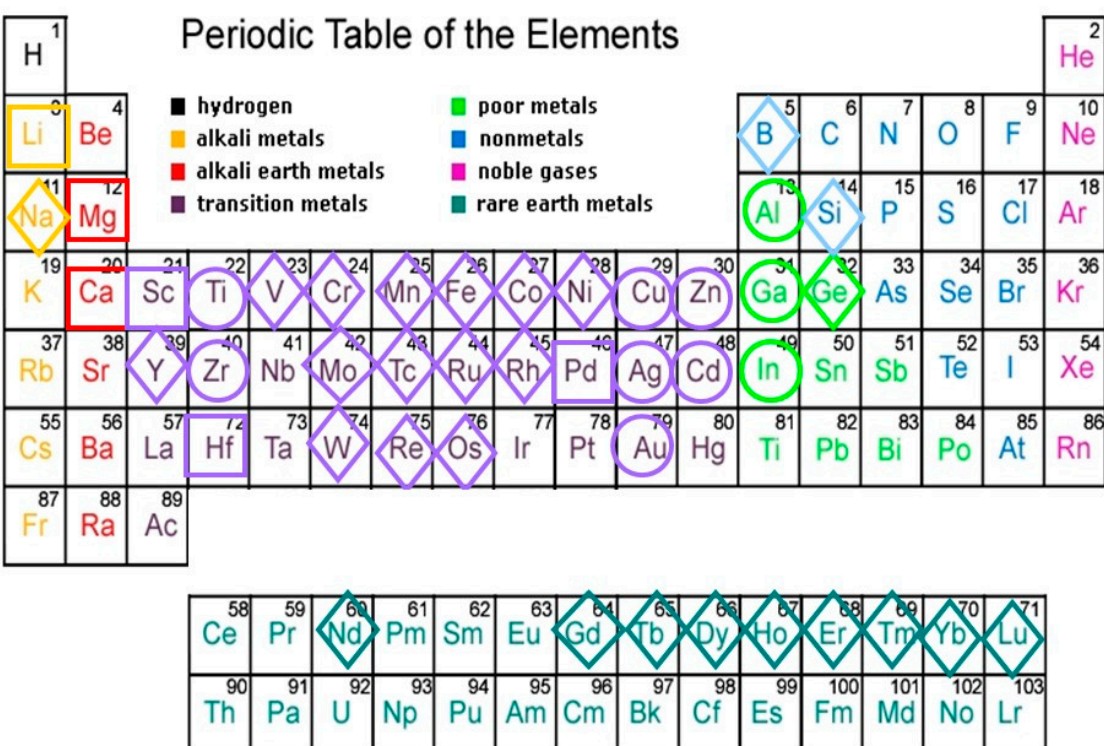

**Figure 5.** Chemical elements found in thermodynamically-stable QC alloys. Main forming elements (Al, Ga, In, Ti, Zr, Zn, Cd, Cu, Ag, and Au) are circled. The second major constituents are squared. Minor constituents are framed within a diamond.

Substitution of minor constituents by next neighbor elements along a given row in the periodic table has been exploited in order to obtain the family of stable quaternary i-QCs given by the formula

$$Al_{70}Pd_{20}\begin{pmatrix} Mn \\ Tc \\ Re \end{pmatrix}_{10} \mapsto Al_{70}Pd_{20}\begin{pmatrix} V \\ Cr \\ Mo \\ W \end{pmatrix}_5 \begin{pmatrix} Co \\ Fe \\ Ru \\ Os \end{pmatrix}_5.$$

The ternary i-QC alloy systems

$$Ag_{42}In_{42}\begin{pmatrix} Yb \\ Ca \end{pmatrix}_{16}, \quad Au_{65-70}\begin{pmatrix} Si \\ Ge \end{pmatrix}_{16-19}\begin{pmatrix} Gd \\ Yb \end{pmatrix}_{14-16},$$

were obtained in a similar vein, the former from the parent binary $Cd_{84}(Ca,Yb)_{16}$ i-QC by replacing Cd atoms by equal amounts of Ag and In ones, which flank Cd element in the periodic table. Finally, it is worth noticing the existence of a great variety of stable i-QCs based on elements belonging to different blocks of the periodic table, but all of them bearing Sc atoms as their second or third constituent, as it is shown in Table 2. Taking into account that Sc atoms have two possible effective valences depending upon whether they are alloyed with non-TM or TM elements, respectively (see Figure 4), we can identify two different possible broad e/a intervals yielding stable QCs, namely, $1.8 \leq e/a \leq 1.9$ and $2.0 \leq e/a \leq 2.2$.

**Table 2.** Sc bearing icosahedral quasicrystals (i-QCs) belonging to different alloy systems grouped by their electron-per-atom e/a ratio value [5,33].

| Compound | e/a |
|:---:|:---:|
| $Zn_{88}Sc_{12}$ | 2.148 |
| $Zn_{71.5}Cu_{12.3}Sc_{16.2}$ | 1.797 |
| $Cu_{46}Al_{38}Sc_{16}$ | 1.817 |
| $Zn_{74}Sc_{16}\begin{pmatrix} Ag \\ Au \end{pmatrix}_{10}$ | 1.823 <br> 1.822 |
| $Zn_{75}Sc_{16}Pd_9$ | 1.829 |
| $Zn_{74}Sc_{16}Ni_{10}$ | 1.838 |
| $Zn_{77}Sc_{16}Fe_7$ | 1.857 |
| $Zn_{78}Sc_{16}Co_6$ | 1.866 |
| $Zn_{74}Sc_{16}Pt_{10}$ | 1.885 |
| $Al_{54}Pd_{30}Sc_{16}$ | 2.126 |
| $Zn_{81}Sc_{15}Mg_4$ | 2.174 |
| $Zn_{77}Sc_8\begin{pmatrix} Ho \\ Er \\ Tm \end{pmatrix}_8 Fe_7$ | 1.991 |
| $Cu_{48}Ga_{34}Sc_{15}Mg_3$ | 2.001 |

The important role played by the presence of a deep DOS minimum close to the Fermi energy in the stabilization of both icosahedral and decagonal QCs [16,29,34] inspired a chemical synthesis exploration project aimed at obtaining new quasicrystalline compounds via pseudogap electronic tuning, which has rendered successful results in the Zn-Sc-Cu, Ca-Au-In, and Mg-Cu-Ga systems [35–38]. Indeed, for a given alloy composition the Fermi-surface–Brillouin-zone interaction mechanism can give rise to the formation of pseudogaps at energy values that are not close enough to the Fermi energy to significantly contribute to reduce its electronic energy. Now, if that pseudogap occurs above the Fermi energy then one can change the alloy's composition by properly alloying the original compound with electron-rich atoms in order to change its e/a ratio, shifting $E_F$ to properly match the Fermi level inside the deeper region of the pseudogap. This strategy was systematically applied to different compounds by Corbett and Lin, starting with $Zn_{17}Sc_3$ alloy (e/a = 2.175) whose structure appeared to be isotypic with that of the prototypic Tsai-type 1/1 approximant (see Section 5). This alloy has an electron concentration ratio larger than that observed in $Cd_{85}(Ca,Yb)_{15}$ i-QCs (e/a = 2.026), but substitution of some Zn by Cu yields a novel ternary icosahedral phase with the stoichiometry $Zn_{71.5}Sc_{16.2}Cu_{12.3}$ (e/a = 2.058).

The electronic tuning via the compositional change route was subsequently applied to $Zn_{11}Mg_2$ alloy precursors to get the quaternary i-$Cu_{48}Ga_{34}Sc_{15}Mg_3$ (e/a = 2.001) phase [36], and the ternary i-$Zn_{82.1}Sc_{14.6}Mg_{3.3}$ (e/a = 2.170) and i-$Au_{44.2}In_{41.7}Ca_{14.1}$ (e/a = 2.15) compounds [35,37]. Remarkably enough, the systematic exploration of the active-metal-gold-post-transition-metal Na-Au-Ga system led to the discovery of the first Na-containing QC, i-$Ga_{37.5}Na_{32.5}Au_{30}$, which has a comparatively low e/a ≃ 1.753 value [38]. However, the existence of the stable phase i-$Au_{62.7}Al_{23.0}Ca_{14.3}$ with the even lower e/a ≃ 1.605 value was reported later on [39].

## 4. Assessing the Quasiperiodic Order Role

There are several hints signaling the importance of short-range effects in the emergence of some unusual physical properties of QCs. For instance, electrical conductivity measurements show that (i) amorphous precursors of i-AlCuFe phase already exhibit an increase of conductivity with temperature, and (ii) the structural evolution from the amorphous to the quasicrystalline state is accompanied by a progressive enhancement of the electronic transport anomalies (Figure 6a) [40,41].

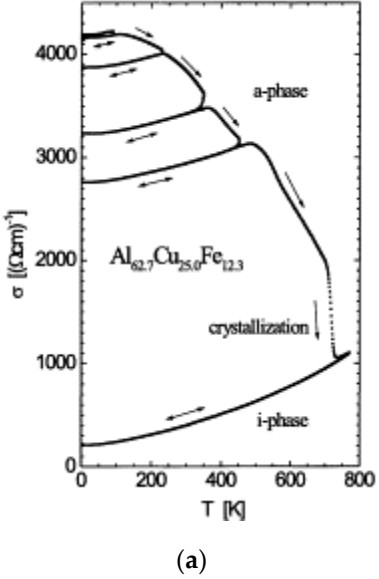
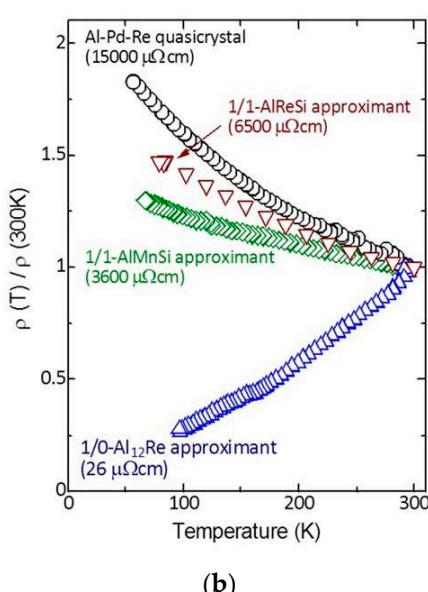

(**a**)    (**b**)

**Figure 6.** (**a**) Temperature dependence of the electrical conductivity of an AlCuFe thin film for different annealing states for the amorphous and for the i-QC phase. The conductivity progressively decreases when the long-range order quality is improved (spanning over more than one order of magnitude at low temperatures) and a clear inverse Matthiessen rule is observed in the σ(T) curves. (Reprinted from Haberken, R.; Khedhri, K.; Madel, C.; Häussler, P. Electronic transport properties of quasicrystalline thin films. *Mater. Sci. Eng. A* **2000**, *294-296*, 475-480. Copyright (2000), with permission from Elsevier); (**b**) temperature dependence of the electrical resistivity of a i-AlPdRe QC along with 1/1-AlReSi, 1/1-AlMnSi, and 1/0-$Al_{12}$Re approximants. The low-temperature electrical resistivity values are indicated in the graph. (Reprinted from Takagiwa, Y.; Kirihara, K. Metallic-covalent bonding conversion and thermoelectric properties of Al-based icosahedral quasicrystals and approximants. *Sci. Technol. Adv. Mater.* **2014**, 15, 044802; doi:10.1088/1468-6996/15/4/044802, Creative Commons Atribution-NonCommercial-ShareAlike 3.0 License).

It is worthy noticing that the σ(T) curves reported for both amorphous and QC phases are almost parallel to each other, hence displaying the inverse Matthiessen rule (see Figure 2a). In an analogous way, crystalline approximants, which exhibit a local atomic environment very similar to their related QC alloys, appear as natural candidates to investigate the relative importance of short-range versus long-range order effects on the transport properties. In fact, many unusual physical properties of QCs are also found in approximant phases, where transport measurements indicate that high-order

approximants (say, 1/1 or 2/1), with lattice parameters exceeding about 2 nm, exhibit similar electrical properties to full-fledged QCs, whereas low-order approximants usually display a typically metallic behavior (Figure 6b). Furthermore, the observed anomalies are generally more pronounced in the case of QCs. These results indicate that the electronic properties of QCs and their related approximant phases are quite similar, provided that the order of the considered approximants is high enough, thereby guaranteeing the local atomic arrangements are essentially the same, irrespective of the long-range order.

On the contrary, transport properties of metallic alloys with complex unit cells, having a similar number of atomic species to those of approximant phases, but not exhibiting the local isomorphism property characteristic of QCs, are typically metallic instead [42]. Therefore, mere structural complexity is not a sufficient condition to give rise to the emergence of anomalous transport properties in QCs. On the other hand, certain anomalous transport properties, such as a high electrical resistivity value or a negative temperature coefficient of electrical resistivity, can also be observed in some crystalline alloys consisting of metallic elements (e.g., Heusler-type $Fe_2VAl$ and $Ru_2TaAl$ alloys) whose structures are unrelated to those of QCs, although both of them share some characteristic features in their electronic structures, namely, a deep and relatively narrow pseudogap close to $E_F$ [43,44]. Taken together, these observations strongly suggest the importance of short-range effects on the emergence of certain non-metallic properties, which are then appreciably intensified, as long-range QPO progressively pervades the overall structure. In turn, it is tempting to assign a chemical nature to these local effects, probably involving the formation of covalent bonds among certain neighboring atoms.

At this stage it is convenient to highlight that the self-similar symmetry characteristic of fully-fledged QCs has the important effect of reproducing the same local patterns once and again at different spatial scales, thereby relaxing the very notion of "local scale". In the mathematical literature this property is referred to as local isomorphism and its significant role in interference processes can be described in terms of the so-called Conway's theorem, which states that distances between identical local arrangements of atoms scale with their own characteristic size [45]. That is, any finite-size region reappears once and again in a non-periodic fashion, but always having slightly different surroundings. Thus, the notion of repetitiveness, typical of periodic arrangements, should be replaced by that of local isomorphism, which expresses the occurrence of any bounded region of the structure infinitely often across the whole volume. In the particular cases of Fibonacci chain and Penrose tiling, Conway's theorem states that given any local pattern having a certain characteristic length, L, at least one identical pattern can be found within a distance of 2L, as it is illustrated in Figure 7.

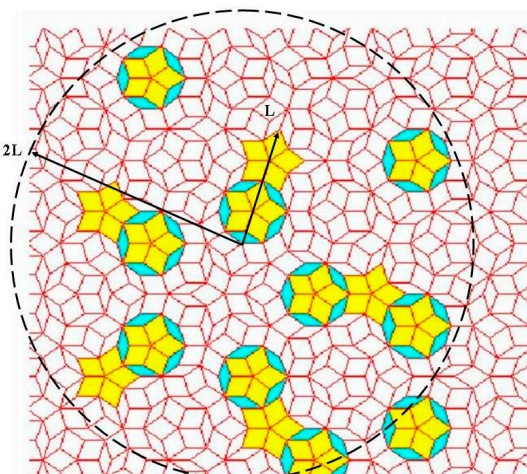

**Figure 7.** Illustration of Conway's theorem in a Penrose tiling: Given an arbitrary pattern of tiles (indicated by L in the picture) one will always find a replica of it at a distance smaller than twice its length (dashed circle).

Therefore, the physical reason for the presence of so many reflections in the diffraction spectra of QCs (see Figure 1a) is a direct consequence of QPO, which guarantees the existence of suitable interference conditions at multiple scales. In the same vein, the existence of critical wave functions with a recurrent spatial pattern in QCs (see Figure 3) is closely related to the competition between local isomorphism and long-range QP order. Thus, the absence of periodicity favors the localization of the wave function in a local pattern. However, according to Conway's theorem this local pattern must have duplicates that extend throughout the whole structure. Then, quantum resonances between such localized states at identical local configurations favor a tunneling effect which gives rise to a hopping mechanism. As a result, the wave functions become more extended in nature, ultimately resulting in electronic and phonon states exhibiting a self-similar structure, in the sense that certain portions of the wave function amplitudes, extending over identical configurations, only differ by a scale factor (see Figure 3b). In summary, the presence of self-similar symmetry in QCs structures renders the difference between short-range and long-range scales, to which we are so used to in the study of periodic structures, somewhat fuzzy, as their overall architecture can be regarded as a hierarchical arrangement of certain basic building blocks at different scales.

## 5. Quasicrystals as Cluster Aggregates

These building units generally adopt well-defined polyhedral shapes and can be described as regular arrangements of atoms in nested shells adopting point group icosahedral symmetries (dodecahedron, icosahedron, triacontahedron, icosidodecahedron). By attending the detailed structure of these polyhedral i-QCs, they can be classified into three main classes, namely: (1) The AlMnSi class, whose prototype is given by the $\alpha$-$Mn_{12}(Al,Si)_{57}$ phase; (2) the MgAlZn class, whose prototype is given by the $Mg_{32}(Al,Zn)_{49}$ Bergman phase; and (3) the CdYb class, whose prototype is given by the $Cd_6Yb$ phase. In all cases the entire self-similar atomic structure can be obtained starting from a given polyhedral seed by systematically applying an inflation process via successive substitutions of atoms by clusters. The inflation operation consists of a proper rescaling by a scale factor $\tau^3$, where $\tau = (1 + \sqrt{5})/2$ is the so-called golden mean. An example of the resulting structure is shown in Figure 8b, displaying an icosidodecahedral aggregate consisting of icosidodecahedral atomic clusters.

As it deduced from X-ray and neutron diffraction data the structure of i-AlPdMn and i-AlCuFe QCs is based on the so-called pseudo-Mackay cluster, which contains 51 atoms and can be described in terms of three centrosymmetric atomic shells: A core of nine atoms, an intermediate icosahedron of 12 atoms, and an external icosidodecahedron of 30 atoms (Figure 8a) [46–49]. The structures of stable i-QCs in the AlCuLi, ZnMgZr, ZnMgRE, ZnMg(Al,Ga) and MgAlPd systems were grouped in the MgAlZn class, which is based on the so-called Bergman clusters: The center is vacant, the first shell is an Al/Zn icosahedron, the second shell is a Mg dodecahedron, and the third shell is a larger Al/Zn icosahedron. Finally, stable i-QCs found in the binary Cd(Yb,Ca) and the ternary ZnScM, (Ag,Au)In(Yb,Ca), and CdMgRE alloy systems have been grouped in the CdYb Tsai-cluster third class, which constitutes the largest of the three classes of i-QCs. In the core of each Tsai-type cluster there is a tetrahedron created by four positionally-disordered Cd atoms, the first shell is a Cd dodecahedron, the second shell is a Yb icosahedron, and the third shell is a Cd icosidodecahedron [50,51].

A major question in the field is whether clusters are physically-significant, chemically-stable entities, or simply convenient geometrical constructions. A number of evidences supporting the existence of clusters as stable physicochemical entities have been reported in the literature, including secondary electron imaging [52], X-ray photoelectron diffraction [53], and scanning tunneling microscopy techniques [54]. The obtained results support the picture of i-QCs as cluster aggregates, which can be described in terms of a three-dimensional QP lattice properly decorated by atomic clusters having the same point symmetry as the whole QC [55]. This scenario naturally brings about some questions, which are the focus of intensive current research. For example, a given cluster may act as a chemically-stable structure when isolated, but it may lose its identity, due to strong interactions with close neighbors, when assembled to form a solid. So, what is the nature of the chemical bonding among the atoms

belonging to a given cluster, as well as among different clusters themselves? Then, along with the stability of clusters we should also consider those aspects related to their reactivity. In a similar way, one may wonder about the number and structure of the different atomic clusters compatible with the system's stoichiometry, or regarding the more appropriate packing rules involving atomic clusters at different hierarchical stages.

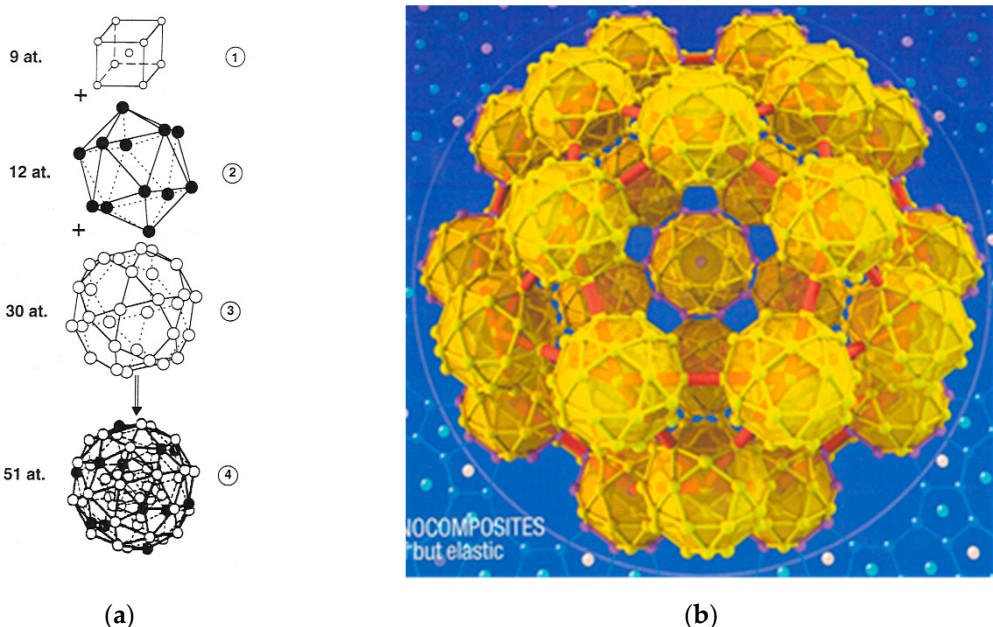

(**a**) 　　 (**b**)

**Figure 8.** (**a**) Structure of a pseudo-Mackay cluster containing an inner cubic shell (1), followed by an icosahedron shell (2), and an icosidodecahedron (3) shell. The number of atoms in each shell is indicated in the figure. White (black) circles denote Al (TM) atoms, respectively. (Reprinted figure with permission from Janot, C. Conductivity in quasicrystals via hierarchically variable-range hopping. *Phys. Rev. B* **1996**, *53*, 181. Copyright (1996) by the American Physical Society); (**b**) A three-dimensional perspective of the $\tau^3$-inflated cluster forming the basic icosidodecahedron structural motif of i-CdYb phase. Its radius is about 12 nm. (Reprinted by permission from Springer Nature: *Nature Materials*, Takakura, H.; Pay-Gómez, C.; Yamamoto, A.; de Boissieu, M.; Tsai, A. P. Atomic structure of the binary icosahedral Yb-Cd quasicrystal. **2007**, *6*, 58-63, Copyright (2007)).

## 6. Chemical Fingerprints in the Physical Properties

An abrupt change in certain physical properties in a series of compounds may indicate an abrupt change in the bond type (e.g., from mainly metallic to mainly covalent). For the sake of illustration, in Table 3 we list the melting temperatures of several i-QCs based on the three known cluster types (Bergman, Mackay, and Tsai). Broadly speaking one would expect relatively low melting points in molecular solids. Thus, the relatively high melting points observed in Al-based QCs may be indicating stronger than expected bonds among neighboring clusters. Alternatively, one may also think of the hierarchical spatial arrangement of clusters as precluding an easy separation from each other, even if they interact weakly among them. In this regard, it is interesting to note that the melting point of the $Al_{65}Cu_{20}Fe_{15}$ cubic approximant alloy is $T_m = 1281$ K, about 12% higher than that of the related icosahedral phase [56]. This may be indicative of long-range QPO effects in QCs.

**Table 3.** Debye ($\Theta_D$) and melting ($T_m$) temperatures of i-QCs belonging to different alloy systems listed by their decreasing melting temperature value.

| Compound | $\Theta_D$ (K) | $T_m$ (K) | Cluster-Type | References |
|---|---|---|---|---|
| $Al_{70.5}Pd_{21}Re_{8.5}$ | 425 | 1298 | Mackay | [57] |
| $Al_{63}Cu_{25}Fe_{12}$ | 509 | 1169 | Mackay | [58] |
| $Al_{63}Cu_{24}Fe_{13}$ | 536 | 1145 | Mackay | [59] |
| $Al_{65}Cu_{20}Fe_{15}$ | 360–560 | 1135–1163 | Mackay | [56,58,60] |
| $Al_{72}Pd_{20}Mn_8$ | 362 | 1140 | Mackay | [61] |
| $Ti_{40}Zr_{40}Ni_{20}$ | 325 | 1083 | Bergman | [62] |
| $Cd_{85.1}Yb_{14.9}$ | 140 | 909 | Tsai | [63] |
| $Al_{60}Li_{30}Cu_{10}$ | 465 | 895 | Bergman | [64] |
| $Au_{44.2}In_{41.7}Ca_{14.1}$ | | 848 | Tsai | [37] |
| $Zn_{88}Sc_{12}$ | | 778 | Tsai | [65] |
| $Zn_{43}Mg_{37}Ga_{20}$ | 240 | 680 | Bergman | [8] |

The electronic transport properties (electrical conductivity, Seebeck and Hall coefficients) of QCs critically depend on minute changes in the structural quality and chemical composition of the considered samples. For instance, high electrical resistivity values can only be obtained after a specific annealing process [56], applied to homogenize the chemical composition and to improve the structural order of the samples. This is a strong indication that a precise chemical composition and a high structural quality are required to attain the high resistivity reported values. It is also interesting to note that QCs belonging to the Mackay-type cluster generally exhibit significantly higher low temperature resistivities and resistivity ratios R = $\varrho$(4 K)/$\varrho$(300 K), than those observed in QCs containing either Bergman- or Tsai-type clusters, which display room temperature resistivity values comprised within the range 100–300 $\mu\Omega$cm, typical of conventional ternary alloys. In a similar way, binary i-(Cd,Ca)Yb QC representatives and the i-AgInYb-related phases, characterized by a high-quality structural QPO with little chemical disorder, exhibit higher electrical conductivities, comparatively large electronic-specific heat coefficients, relatively low Debye temperatures, and higher magnetoresistance values than those reported for Al-based QCs [9,33]. This remarkable behavior, combined with the relevant role of p-d hybridization effects in the emergence of a pseudogap close to the Fermi level, certainly points to a significant influence of chemical effects in the transport properties of this QC alloy system [66,67].

Several chemical trends are also observed in the transport properties of i-QCs belonging to the AlCu(Fe,Ru,Os) and AlPd(Mn,Re) systems. This is illustrated in Table 4, where we clearly see a systematic increase of both the low temperature resistivity and the resistivity ratio maximum values by increasing the Z number of the group 7 (group 8) third element, respectively. Thus, QCs containing heavier TM atoms, such as AlPdRe and AlCuOs, possess higher resistivity and R ratio values than those observed in AlPdMn and AlCu(Fe,Ru) phases, the i-AlPdRe being the most resistive QC discovered so far, exhibiting maximum $\varrho$(4 K) and R values two orders of magnitude larger than those of the isostructural i-AlPdMn phase. Such a resistivity behavior was interpreted as signaling an enhancement of covalent bonding between Al and TM atoms in AlPdRe i-QCs [68] (see Section 7). By inspecting the two last rows of Table 4, one would expect remarkably low electrical conductivity values in QCs belonging to the quaternary system i-AlPdOsRe. However, a room temperature resistivity $\varrho = 1.2 \times 10^4$ $\mu\Omega$cm and a resistivity ratio R = 5 was reported for i-$Al_{71.5}Pd_{19}Os_{5.5}Re_4$ [69]. These figures are well below those listed for i-AlPdRe phases in Table 4, indicating the peculiar role of rhenium atoms in icosahedral phases.

**Table 4.** Chemical trends in the low-temperature electrical resistivity ($\varrho$) and the electrical resistivity ratio (R) defined as R = $\varrho$(4 K)/$\varrho$(300 K) of Al-based i-QCs [57,69–75].

| QC System | $\varrho$(4 K) (m$\Omega$cm) | R |
|---|---|---|
| AlLiCu | 0.87 | 1.1 |
| AlCuFe | 2.6–11 | 1.2–2.5 |
| AlPdMn | 1.5–10 | 1.1–2.5 |
| AlCuRu | 1.7–51 | 1.2–4.2 |
| AlCuOs | 77–143 | 4.0–4.5 |
| AlPdRe | 2–2000 | 1.8–280 |

The electrical resistivity trends illustrated in Table 4 may arise from the relativistic contraction of the s and p states relative to the d and f ones, which lowers the orbital energies of the former and screens the nucleus effective charge, so that the outer d electrons experience a lesser binding, hence displaying a larger spatial extent. Accordingly, relativistic effects lead to a lowering of the energy of the s and p bands, along with a simultaneous raising of the energy of the d bands. This enhances sp-d hybridization among these bands, ultimately leading to an increase of cohesive energy and a localization of electrons involved in the formation of covalent bonds.

In Figure 9a we plot the room temperature electrical resistivity for samples belonging to the i-AlCuRu system with a constant aluminum content but different Cu and Ru concentrations. By inspecting this figure we readily appreciate that the $\varrho$(300 K) value systematically increases as we increase the number of electrons per atom. A similar enhancement of the resistivity with increasing concentration of TM atoms was reported in the i-AlPdRe system as well [68]. This is a quite unexpected behavior if we think of it in terms of the free electron model, but it can be better understood if we consider that most electrons are mainly employed in chemical bonding instead, hence becoming essentially localized, rather than extended in character. Indeed, from the crossing among the various spectral electronic distribution curves shown in Figure 9b, one realizes that a substantial electronic overlapping exists between Al 3p and Ru 4d states close to the Fermi level, as well as between Cu 3d and Al 3s,d states in the middle of the band. In particular, the Al–Ru interaction shifts the Al states far from $E_F$, thereby deepening the pseudogap [78–80].

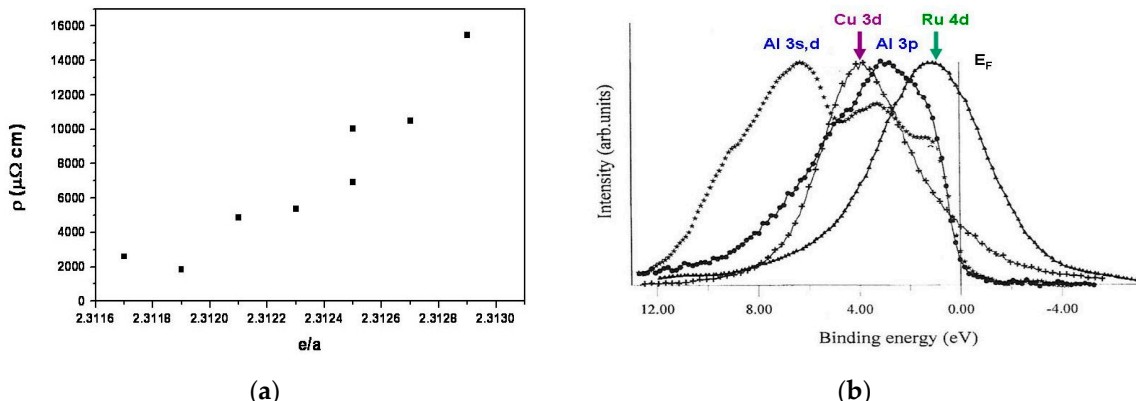

(**a**)　　　　　　　　　　　　　(**b**)

**Figure 9.** (**a**) Room temperature electronic resistivity values retrieved from the literature [14,76,77] as a function of the e/a ratio for i-$Al_{65}Cu_{35-x}Ru_x$ QCs. (**b**) Occupied state distribution in i-$Al_{65}Cu_{20}Ru_{15}$: Al 3p (dotted line), Al 3s,d (starred line), Cu 3d (line with crosses), and Ru 4d (line with triangles) (courtesy Esther Belin-Ferré).

## 7. From Metallic to Covalent Bonding Styles

Some time ago, it was pointed out that the presence of a pronounced hollow feature in the DOS, due to hybridization of a narrow d band with a broad sp band, may be regarded as a common feature in transition-metal aluminides. Whether these hollows become actual gaps, as it was predicted for $RuAl_2$,

FeAl$_2$, and Fe$_2$NbAl alloys [18,81], will depend on fine details of the electronic structure, including the relative positions of the atomic levels, the size of the atoms, the crystal structure, and the number of valence electrons. These results inspired the search for semiconducting QCs, and the electronic structure of several hypothetical approximants corresponding to Al-TM i-QCs was studied by using density functional calculations. In this way, the possible existence of semiconducting approximants and even band insulators in Al-based QCs was put forward [82–84]. These phases should exhibit a QP chemical bond network extending over long distances, which could only be established on an underlying high structural QP arrangement of atoms throughout the space.

Now, since the main building blocks of i-QCs are assumed to be hierarchically-arranged atomic clusters one must consider bonds among atoms inside clusters along with cluster–cluster interactions giving rise to the extended chemical network. The bonding nature of icosahedral clusters of the group 13 elements was earlier investigated to understand the relation between properties of clusters and their own aggregates. By means of molecular orbital quantum calculations it was found that Al$_{12}$ and Al$_{13}$ atomic icosahedra (the latter with one atom located at the center) exhibited covalent and metallic-type bonding, respectively (see Figure 10a, top panel). Thus, atomic occupation of the icosahedron center induces a metallic-covalent bonding conversion phenomenon in these monoatomic clusters [85].

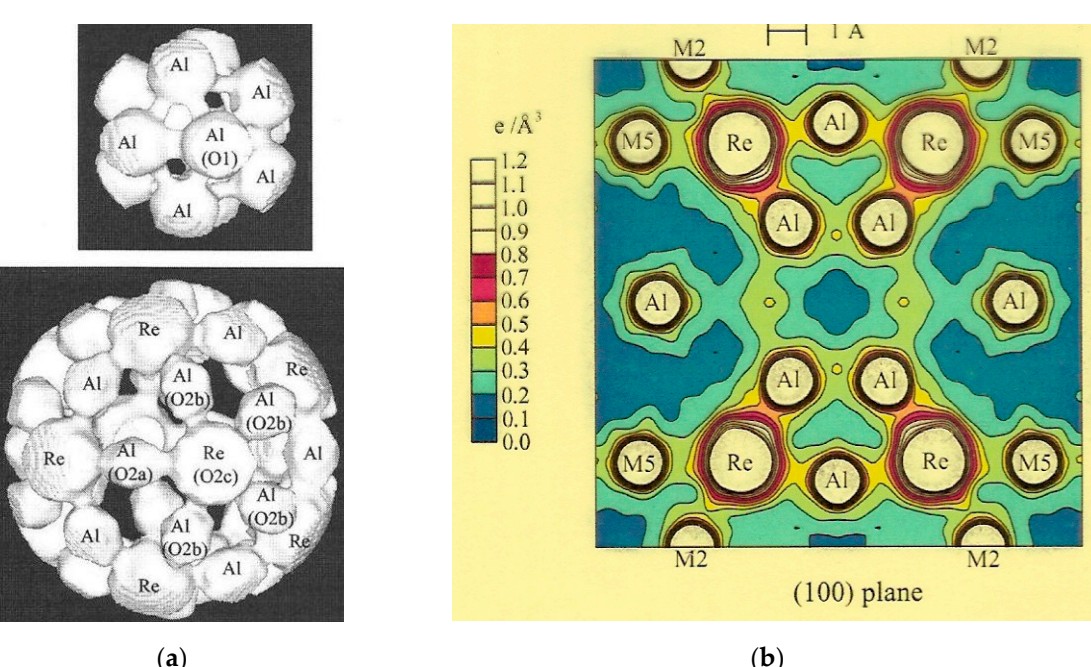

(**a**)　　　　　　　　　　　　　　　　　　　　　　　　　(**b**)

**Figure 10.** (**a**) Electronic charge equidensity surfaces (0.35 e/Å$^3$) of the Mackay icosahedral cluster first (top panel) and second (bottom panel) shells in the α-AlReSi approximant crystal; (**b**) section contour map of charge density of α-AlReSi in the range 0.00–1.20 e/Å$^3$, with a step of 0.10 e/Å$^3$. (Reprinted figure with permission from Kirihara, K.; Nakata, T.; Kimura, K.; Kato, K.; Takata, M.; Kubota, Y.; Nishibori, E.; Sakata, M. Covalent bonds and their crucial effects on pseudogap formation in α-Al(Mn,Re)Si icosahedral quasicrystalline approximant. *Phys. Rev. B* **2003**, *68*, 014205 Copyright (2003) by the American Physical Society).

When going from a metallic to a covalent bonding scenario one may expect to observe some general features in the atomic arrangement of the system, such as a smaller coordination number around most atoms, a sharper directionality of the electronic charge distribution among them, as well as shorter interatomic distances. Indeed, it is well known that different bonding styles (i.e., ionic, covalent, metallic, Van der Waals) are characterized by different values of their corresponding effective radii. Accordingly, it was found that the distance between neighboring Al atoms in Al-based i-QCs is 0.24 nm, approximately 10% shorter than that of fcc-type Al elemental crystals (0.286 nm). On

the other hand, detailed X-ray diffraction studies indicated that the quasilattice constant of i-AlPdRe QCs slightly increases when the TM concentration is increased in the sample, whereas their average atomic radius decreases ($r_{Al}$ = 1.45 Å, $r_{Pd}$ = $r_{Re}$ = 1.37 Å), so that the atomic density (measured in atoms per $nm^3$) decreases due to the related quasilattice expansion. However, the atomic density of the samples, determined from their bulk density (measured in $gcm^{-3}$) divided by the average atomic weight, decreases more rapidly than what is estimated from the quasilattice expansion. This result was interpreted as resulting from the presence of a covalent bonding network involving Al and TM atoms in AlPdRe i-QCs [68].

Indeed, studies of the electronic charge density in the approximant crystals alloys $\alpha$-AlMnSi and $\alpha$-AlReSi (containing Mackay-type clusters) disclosed a strong directional Al-Mn (Al-Re) covalent bond in the second shell of the Mackay cluster (see Figure 10a, bottom panel), with similar bonds existing between the Mn atom in the second shell and the so-called Al glue atoms connecting neighbor clusters (labeled M2 and M5 in Figure 10b). In this way, the Al-TM bonds in the second shell are considered to play an important role in the stabilization of the corresponding clusters [17,86]. In addition, the interatomic covalent bonds of the $\alpha$-AlReSi approximant are stronger than those of the $\alpha$-AlMnSi one. In this way, the concept of metallic-covalent bonding conversion was proposed by Kimura and co-workers in order to describe a local phenomenon involving most valence electrons, at variance with the well-known metal-insulator transition which affects the whole solid but it is determined by the electron states close to the Fermi energy only [20,83].

## 8. Potential of Quasicrystals as Thermoelectric Materials

As we have discussed in the previous sections, the electronic and thermal transport properties of stable i-QCs exhibit a more semiconductor-like than metallic character (see Table 1). Quite interestingly from the viewpoint of possible practical applications, the temperature dependence of several transport coefficients in QCs suggest that these compounds may be promising thermoelectric materials (TEMs) [20,87]. In fact, the efficiency of a thermoelectric device is measured in terms of the so-called thermoelectric figure of merit, which is defined by the expression

$$ZT = \frac{\sigma\, S^2\, T}{\kappa} \tag{1}$$

where σ(T) is the electrical conductivity, S(T) is the Seebeck coefficient, and κ(T) is the thermal conductivity [88]. According to Equation (1) there are four basic reasons supporting QCs and their approximants as potential thermoelectric materials (TEMs) [87]:

1. The electrical conductivity of i-QCs steadily increases as the temperature increases up to the melting point, as it is illustrated in Figures 2 and 11a.

2. Al-bearing i-QCs show relatively large Seebeck coefficient values (50–120 $\mu VK^{-1}$) as compared to those observed in usual metallic alloys (1–10 $\mu VK^{-1}$) at room temperature, and the S(T) curves usually deviate from the linear behavior characteristic of charge diffusion in ordinary metallic alloys in the temperature interval 100–300 K (Figure 11b).

3. Furthermore, in the case of i-AlCu(Fe,Ru,Os) and i-AlPd(Mn,Re) representatives, both σ(T) and S(T) increase as the temperature is increased over a broad temperature range (T = 100–550 K), which yields relatively high power factor ($\sigma S^2$) values [89].

4. The thermal conductivity of most i-QCs (within the range $\kappa$ = 1–5 $Wm^{-1}K^{-1}$ at room temperature) is about two orders of magnitude lower than that of ternary alloys, since it is mainly determined by the lattice phonons instead of charge carriers over a wide temperature range (Figure 11c). These unusually-low thermal conductivity values are comparable to those observed for thermal insulators of extensive use in aeronautical industry [3].

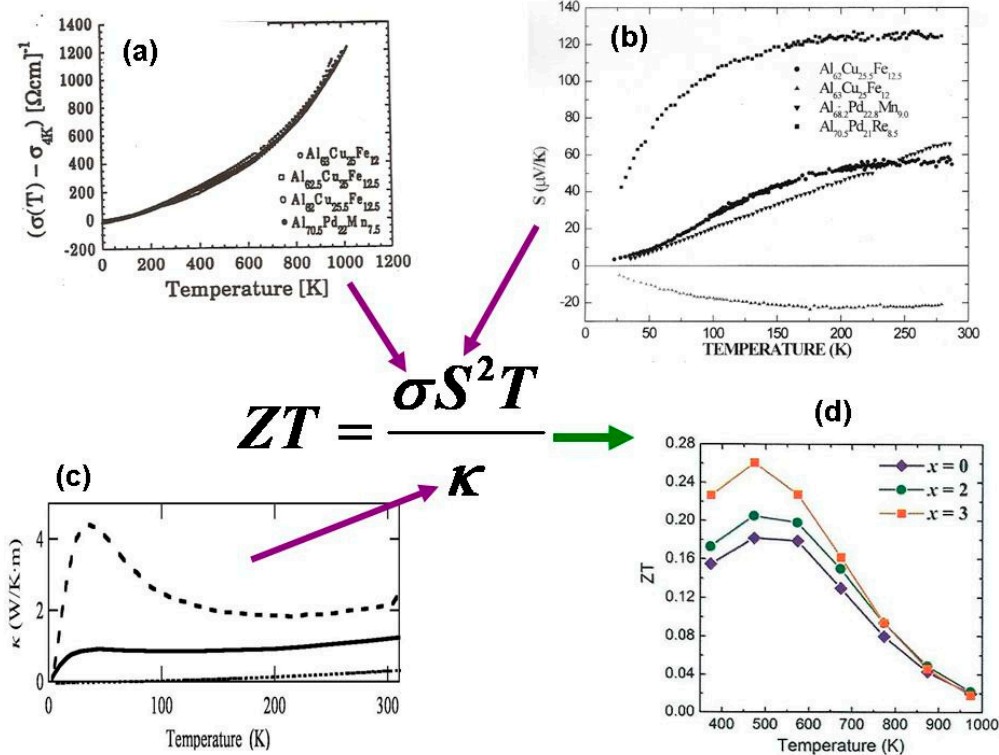

**Figure 11.** Collage picture illustrating the contribution of each transport coefficient (magenta arrows) to the resulting ZT value (green arrow). Note that panels (**a**–**d**) correspond to different i-QCs representatives, each one showing an optimal behavior for the considered physical magnitude: (**a**) Temperature dependence of the electrical conductivity for four different i-QCs up to 1000 K. (Reprinted figure with permission from Mayou, D.; Berger, C.; Cyrot-Lackmann, F.; Klein, T.; Lanco, P. Evidence for unconventional electronic transport in quasicrystals. *Phys. Rev. Lett.* 1993, 70, 3915-3918. Copyright (1993) by the American Physical Society); (**b**) temperature dependence of the Seebeck coefficient for different i-QCs. (Courtesy of Roberto Escudero); (**c**) temperature dependence of the thermal conductivity for the i-$Al_{74.6}Re_{17.4}Si_8$ QC (solid line) and its 1/1-cubic approximant (dashed line). The contribution due to the electrons (estimated from the Wiedemann–Franz law, dotted line) is almost negligible as compared to that due to the phonons (Courtesy Tsunehiro Takeuchi); (**d**) figure of merit (ZT) curves as a function of temperature for $Al_{71-x}Ga_xPd_{20}Mn_9$ (x = 0, 2, 3). (Reprinted from Takagiwa, Y.; Kirihara, K. Metallic-covalent bonding conversion and thermoelectric properties of Al-based icosahedral quasicrystals and approximants. Sci. Technol. Adv. Mater. 2014, 15, 044802; doi:10.1088/1468-6996/15/4/044802, Creative Commons Atribution-NonCommercial-ShareAlike 3.0 License).

The low thermal conductivity of QCs can be understood in terms of two main facts. For one thing, heat propagates mainly by means of phonons, since the charge carrier concentration is severely reduced due to the presence of a pseudogap around the Fermi level (see Section 2). On the other hand, reciprocal space has a nearly fractal pattern in QCs (see Figure 1a), so that the transfer of momentum to the lattice is not bounded below, and the thermal current intensity is strongly reduced due to an enhancement of phonon–phonon scattering processes occurring at all scales in the reciprocal space. In addition, the high fragmentation of the frequency spectra in QCs leads to a small group velocity of phonons. In a similar way, an increased number of Umklapp electron–phonon processes in QCs may enhance the phonon drag contribution to the total Seebeck coefficient as well [90].

Inspired by these features a series of experimental studies aimed at assessing the possible use of i-QCs belonging to different alloy systems as potential TEMs has been performed by several research groups. In this way, the most promising QC alloy system was first determined by searching for

that exhibiting the lowest thermal conductivity value. Then, the high sensitivity of QCs transport coefficients to minor stoichiometric changes was exploited in order to properly enhance the power factor of the selected QCs, without seriously compromising their characteristic low thermal conductivities.

In Table 5 we list the transport coefficients for those representatives yielding the best ZT values at room temperature. By inspecting Table 5 we see that isostructural i-$Al_{71}Pd_{20}Mn_9$ and i-$Al_{71}Pd_{20}Re_9$ samples exhibit promising ZT values, which are comparable to those reported for half-Heusler, skutterudites, and clathrates compounds at room temperature. In this way, it was found that the more promising QCs belong to the AlPd(Mn,Re) system. Subsequent search was then focused on refining their stoichiometries to further increase the corresponding ZT values. Indeed, ZT values differing by more than two orders of magnitude can be attained in a single QC system by slightly changing the sample's composition by a few atomic percent (hence preserving the QP lattice structure). We also note that both positive and negative values of the Seebeck coefficient can be obtained in this way, which allows for both the n- and p-type legs present in a typical thermoelectric cell to be fabricated from the same material. Furthermore, enhanced ZT values (0.2–0.25) are obtained at higher temperatures (450–500 K) for related quaternary i-QCs, as it is shown in Figure 11d for i-$Al_{71-x}Ga_xPd_{20}Mn_9$ representatives [91]. The main reason for this ZT enhancement is the lowering of phonon thermal conductivity by the weakening of the inter-cluster bonds along with the usual alloying effects [92].

**Table 5.** Room temperature values of the electrical conductivity ($\sigma$), Seebeck (S), thermal conductivity ($\kappa$) transport coefficients and figure of merit (ZT) for i-QCs belonging to different alloy systems.

| Compound | $\sigma$ $(\Omega cm)^{-1}$ | S $(\mu VK^{-1})$ | $\kappa$ $(Wm^{-1}K^{-1})$ | ZT | Reference |
|---|---|---|---|---|---|
| $Zn_{57}Mg_{34}Er_9$ | 6170 | +7 | 4.5 | 0.002 | [93] |
| $Al_{65}Cu_{20}Ru_{15}$ | 250 | +27 | 1.8 [2] | 0.003 | [76] |
| $Cd_{84}Yb_{16}$ | 5560 | +14 | 9.4 | 0.004 | [94] |
| $Ag_{42.5}In_{42.5}Yb_{15}$ | 5140 | +12 | 4.8 | 0.005 | [95] |
| $Al_{62.5}Cu_{24.5}Fe_{13}$ | 310 | +44 | 1.8 [1] | 0.010 | [76] |
| $Al_{64}Cu_{20}Ru_{15}Si_1$ | 390 | +50 | 1.8 [2] | 0.020 | [76] |
| $Al_{71}Pd_{20}Re_9$ | 450 | +80 | 1.3 | 0.070 | [91] |
| $Al_{71}Pd_{20}Mn_9$ | 714 | +90 | 1.5 | 0.120 | [91] |

[1] From reference [96]. [2] Estimated upper limit.

At this point; however, we must highlight a main difference between QCs and other TEMs, such as skutterudites and clathrates, exhibiting similar ZT values at room temperature. This difference refers to the rapid increase of the thermal conductivity with the temperature in Al-based i-QCs (and related approximants) starting at temperatures above T ~300−400 K [97–99]. By all indications this increase mainly arises from the electronic contribution, while the phonon contribution steadily decreases as T increases [97–102]. This $\kappa$(T) increase leads to ZT curves which progressively decrease as the temperature is raised above the Debye temperature ($\approx$ 450–500 K), as can be seen in Figure 11d. Conversely, the ZT(T) curves of both skutterudites and clathrates progressively increase within this temperature range, attaining high figure of merit values (within the interval ZT = 0.8–1.3) at temperatures around 800–900 K. The enhancement of the electronic contribution to the thermal conductivity in QCs may be accounted for, within the framework of a two-band electronic structure model, in terms of a bipolar diffusion term [98], along with a possible dependence of the charge carrier density with the temperature at high enough temperatures [103]. Therefore, the high-temperature ZT value could be improved if the pseudogap width was widened in order to reduce the unfavorable contribution related to bipolar transport effects. Quite remarkably; however, although it has been reported that the increase of the thermal conductivity of QCs with temperature reduces their potential as competitive TEMs at both moderate and high temperature regimes, this unusual feature makes them attractive for the design of thermal rectifiers [99].

### 9. Outlook

Electron density maps derived from X-ray synchrotron analysis performed in approximant crystals provide compelling evidence for the existence of covalent bonds involving Al and TM atoms in the unit cell. These results support the picture of a covalent-metallic bonding network extending through the structure of these approximant crystals as well as their related QCs phases, ultimately leading to a relatively-weak structure of rigid cluster aggregates. Indeed, the physical role of chemically-stable atomic clusters in the QC's architecture has been recently supported by detailed studies on the solidification dynamics of d-QCs and related approximants in the Al-Co-Ni alloy system, showing that the growth of the solid phases proceeds via attachment of relatively-large atomic clusters from the liquid, rather than by the direct assembly of individual atoms from the melt [104,105].

Nevertheless, electronic structures of i-QCs and related approximants have barely been understood on the basis of the molecular orbitals of their constituent clusters [82]. Thus, rather than trying to explain the specific properties of QCs in terms of the conceptual schemes originally introduced to describe purely metallic, semi-metallic, semiconductor, or insulator behaviors, it may be advisable to introduce a broader perspective, able to properly blend typical properties of both metallic and covalent solids in terms of the main features observed in the electronic structure of QCs. Accordingly, the precise relationship between the Fermi-surface–Brillouin-zone mechanism, sp-d hybridization, and covalent bonding formation should be further clarified in order to gain a proper understanding of the bond nature in QCs [106]. In this regard, the visualization of the bonding nature in terms of the crystal orbital Hamilton population will be a very convenient tool [107,108].

In this way, a suitable scenario accounting for the presence of a deep pseudogap close to the Fermi level could be envisioned as a two-step process where the Fermi-surface–Brillouin-zone mechanism efficiently removes extended electronic states from higher to lower energy values around $E_F$ value; then some of these lower energy states are able to hybridize with states belonging to low-lying d-orbitals, giving rise to the formation of bonding (antibonding) states below (above) $E_F$, further increasing the stability of the structure through the formation of an extended network of covalent bonds arranged according to the characteristic long-range QPO of fully-fledged QCs. According to this picture in the pseudogap region, both localized and extended electronic states can be simultaneously present, though most of the remaining states close to the Fermi energy would be localized in character, thereby explaining the reported large low-temperature electrical resistivity values.

Finally, regarding the potential use of QCs as suitable TEMs, it is currently agreed that some minor improvement in the thermoelectric performance of Al-based i-QCs in the temperature interval 300–400 K may be obtained by a judicious choice of both sample composition (i.e., via a fourth element alloying) and processing conditions via trial-and-error procedures. The simultaneous increase of both $\sigma(T)$ and $S(T)$ transport coefficients with the temperature (a behavior rarely observed in most materials) suggests that ZT could be further improved in the high temperature regime (via power factor enhancement). For instance, a maximum value $\sigma S^2 = 230$ $\mu Wm^{-1}K^{-2}$ at 873 K was measured in the 1/1-$Au_{48}Al_{37}Yb_{15}$ approximant [109]. This figure is significantly higher than the values previously obtained for AlGaPdMn i-QCs exhibiting relatively-high ZT values at lower temperatures (Figure 11d). Nevertheless, the remarkable parallel increase of the thermal conductivity counterbalances the ZT numerator contribution, leading to an overall decrease of the thermoelectric performance of QCs at temperatures above $\Theta_D$. Thus, in order to attain a substantial improvement, it seems convenient to adopt a more direct approach based on an electronic structure engineering strategy [110]. To this end, a deeper understanding on the electronic structure of QCs and the physical basis of their unusual transport properties is convenient. For instance, the construction of a minimal model to discuss the effects of crystalline electronic fields and on-site Coulomb repulsion between the 4f and 5d orbitals at Yb atoms, in a class of Yb-based QCs (i-$Au_{51}Al_{34}Yb_{15}$) and related approximant crystals, has recently been reported to account for the intermediate valence state of Yb atoms in this alloy system [111].

In fact, since the ZT value depends on the charge carrier concentration, in addition to control the de-doping level via suitable chemical substitution [112,113], one may exploit the existence of multiple

degenerate bands close to the Fermi level, arising from the high-order symmetry of i-QCs in the reciprocal space. In this way, one may think of an electronic structure exhibiting two bands possessing a different energy-width and overlapping with each other close to the Fermi level as a promising scenario for efficient TEMs [114]. To this end, the DOS enhancements stemming from hybridization of the Yb f-states and Au/Al sp-states in both i-QCs, and related approximants belonging to the Au-Al-Yb alloy system, may play a significant role. These representatives that, along with the binary i-Cd-RE (RE = Gd to Tm, Y) QCs, have recently received growing attention due to their interesting magnetic properties [115–117] could; thus, bring in some promising news to the thermoelectric materials research community in the years to come.

**Funding:** This research received no external funding.

**Acknowledgments:** This work is dedicated to the memory of Esther Belin-Ferré, with my warm gratitude for her support and continued interest in my research activities during the last two decades. I gratefully thank Uichiro Mizutani, Kaoru Kimura, and Tsunehiro Takeuchi for sharing useful comments and materials and for their continued interest in my work on thermoelectric properties of quasicrystals over the years. I thank Victoria Hernández for the critical reading of the manuscript and Laura Lao for her careful editorial assistance.

**Conflicts of Interest:** The author declares no conflict of interest.

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
