# Peer review of "Chemical Bonding and Physical Properties in Quasicrystals and Their Related Approximant Phases: Known Facts and Current Perspectives"

_applsci, doi:10.3390/app9102132_

Round 1

Reviewer 1 Report

This is very good story on Quasicrystals. The author reviewed a few experimental results and numerical simulations and proposed two main features, including the formation of covalent bonds among certain atoms and the self-similar symmetry of the underlying structure. Finally, the author discussed the potential of quasicrystals as thermoelectric materials on the basis of their specific transport properties. This is a good mini review. I suggest to publish in the current form. 

Author Response

I thank the reviewer for his/her interest in my manuscript and for his/her favorable recommendation for the publication of this work.

Reviewer 2 Report

This review paper covers broad topics of quasicrystal research, i.e., growth, synthesis, transport properties, the Hume-Rothery rule, quasiperiodicity, chemical bonding, and application. I think this paper is suited for publication in Applied Sciences. The following concerns should be considered below before publication.

(1) The title should be replaced because this review paper deals not only chemical bonding but various features in quasicrystals and approximant crystals.

(2) Can the author add the recent active research on magnetic properties for Tsai-type quasicrystals and approximants? Such new topics also attract readers.

(3) I think the present perspectives by the author look weak for chemical bonding nature and application of quasicrystals.

(4) The author should recheck the references.

Author Response

I thank the reviewer for his/her interest in my manuscript and for his/her useful comments and suggestions for the benefit of the readers. Accordingly:

1. I have modified the title according to the suggestions given by the reviewer.

2. I have briefly commented on some recent results on the magnetic properties for Tsai-type QCs and related approximants in the last paragraph of Section 9 (lines 763-766).

3. I have rewritten Section 9 in order to highlight the role of chemical bonding in some potential applications of QCs.

4. I have checked the reference list.

Reviewer 3 Report

The paper reviews the progress of quasicrystals. Several aspects are discussed, including the structures, synthesis method, and the advantage for thermoelectric application. This is a timely topic, which can warrant a review paper. The results are very novel and might be very interesting for the scientific community. My only minor remark concerns are

1. In Figure 11, the data from a, b, c and d were from different references. However, the ZT equation and the corresponding arrows indicate that these data should be from the same sample. Please revise this figure to avoid the misleading.

2. The priority conduction to pursue high thermoelectric performance is to optimize the carrier concentration. Please include some methods to optimize carrier concentration for quasicrystals.

3. At the temperature over 300 K, the bipolar conduction is quite strong, leading to high thermal conductivity at high temperature. How to suppress bipolar conduction in quasicrystals?

Author Response

I thank the reviewer for his/her interest in my manuscript and for his/her useful comments and suggestions for the benefit of the readers. Accordingly:

1. A caution note has been included in the Figure 11 caption (lines 620-622) in order to avoid possible misunderstandings to the readers.

2. I have suggested a couple of methods to optimize the charge carrier concentration in QCs In the last paragraph of Section 9 (lines 756-763).

3. A full paragraph commenting on the behavior of thermal conductivity at temperatures above the room temperature, and its influence on the resulting thermoelectric figure of merit, has been added at the end of Section 8 (lines 670-694). Accordingly, the thermoelectric performance of these alloys in the moderate and high temperature regimes cannot compete with those reported for other TEMs of interest, such as skutterudites and clathrates. To the best of my knowledge, no detailed physical mechanism able to suppress the rapid increase of thermal conductivity with temperature in i-QCs has yet been proposed, though a suitable widening of the pseudogap width will probably work as a first approximation. Fortunately enough this unfavorable feature (from the thermoelectric materials viewpoint) can be usefully exploited in order to design thermal rectifiers.

Round 2

Reviewer 2 Report

I think the revised manuscript is now suited for publication.